# Dietary Intake of Polyphenols or Polyunsaturated Fatty Acids and Its Relationship with Metabolic and Inflammatory State in Patients with Type 2 Diabetes Mellitus

**DOI:** 10.3390/nu14051083

**Published:** 2022-03-04

**Authors:** Marcin Kosmalski, Anna Pękala-Wojciechowska, Agnieszka Sut, Tadeusz Pietras, Bogusława Luzak

**Affiliations:** 1Department of Clinical Pharmacology, Medical University of Lodz, Kopcińskiego 22, 90-153 Łódź, Poland; anna.pekala-wojciechowska@umed.lodz.pl (A.P.-W.); tadeusz.pietras@umed.lodz.pl (T.P.); 2Department of Haemostasis and Haemostatic Disorders, Medical University of Lodz, Mazowiecka 6/8, 92-235 Łódź, Poland; agnieszka.sut@stud.umed.lodz.pl

**Keywords:** type 2 diabetes mellitus, diet, polyphenols, omega-3, inflammation

## Abstract

Background: The aim of the study was to evaluate the relationship between polyphenol or polyunsaturated fatty acids (PUFAs) consumption and the selected metabolic and inflammatory markers in type 2 diabetes (T2DM) patients. Methods: The study enrolled 129 diabetics (49 men, mean age 64.1 ± 9.8 years) with different amounts of polyphenol and PUFAs consumption. Results: A significant effect of polyphenol or PUFAs omega-3 consumption on fasting glucose concentration (FG) or glycated haemoglobin fraction (HbA1c) was reported. A negative association was observed between FG and total polyphenol, flavonoid, flavan-3-ol and stilbene intake. In the group with high flavonoid intake, the FG was significantly lower compared to the group characterised by low flavonoid intake. Polyphenols, except stilbenes, did not modulate HbA1c. Additionally, higher consumption of PUFAs omega-3 significantly decreased HbA1c, and the intake of eicosapentaenoic (EPA) and docosahexaenoic (DHA) acids negatively and significantly correlated with FG and HbA1c. Further analysis confirmed a significant association between EPA + DHA intake and HbA1c, with significant interactions with age and gender or with body mass index and waist-to-hip ratio. The dietary intake of polyphenols or PUFAs was independent of familial diabetes or diabetic diet application. Conclusions: Our study indicates a positive effect of high consumption of flavonoids, omega-3 PUFAs and stilbenes on the markers of carbohydrate metabolism balance and the absence of such an effect on other cardiometabolic markers and inflammatory conditions.

## 1. Introduction

Type 2 diabetes (T2DM) is one of the most common metabolic diseases in the world, and its development is primarily associated with insulin resistance (IR) of peripheral tissues and insufficient insulin production by pancreatic beta cells [1]. Chronic hyperglycaemia and IR lead to dangerous consequences, including microvascular complications (nephropathy, neuropathy, retinopathy and sexual dysfunction) and macrovascular ones (coronary heart disease, cardiomyopathy, arrhythmias, peripheral artery disease, cerebrovascular disease and sudden death) [2,3]. The interaction between environmental and genetic factors lies at the root of T2DM and its complications. Obesity, energy-dense “Western diet”, elderly age and lack of physical activity are the main risk factors for the development of this pathology, and hundreds of genomic loci and variants associated with the pathophysiology of this disease and the mechanism of development of its complications have been discovered by previously conducted research [4,5]. In addition, a link between IR, T2DM and the occurrence of low-intensity systemic inflammation called “metainflammation” has been suggested. This correlation seems to be bidirectional. On the one hand, long-term activation of the innate immune system disrupts the secretion and action of insulin; on the other hand, the hyperglycaemic environment leads to dysfunction of the immune cells [6,7,8].

The basis of T2DM therapy is a diet that should be based on a variety of nutrient-rich foods consumed in the right proportions. Its purpose is to improve the overall health condition, achieve and maintain a healthy body weight, daily glycaemic values, blood pressure and lipids and prevent chronic complications of the disease [9]. Based on the research conducted so far, it has been suggested that the type of diet and food products consumed can affect the metabolic status of the body not only directly but also indirectly by affecting the immune system and the functionality of many organs, including the adipose tissue, liver, pancreatic islets, muscles and brain [10,11]. The available clinical data indicate significant benefits from various “health-promoting” diets, with the Mediterranean diet (MedDiet) recommended for T2DM patients. These diets are characterised, among others, by low meat consumption, the use of oils instead of fats, moderate amounts of red wine and significant amounts of fresh fruit and vegetables. The particular benefits of this diet are associated with the high content of polyphenols and omega-3 polyunsaturated fatty acids (PUFAs omega-3) [12,13,14].

Polyphenolic compounds are a heterogeneous group of molecules with various chemical structures that are secondary metabolites of plants. These include mainly flavonoids (flavan-3-ols, flavones, flavonols, flavanones, dihydroflavonols, anthocyanins, isoflavones and chalcones) and non-flavonoids, including phenolic alcohols, phenolic acids and derivatives, stilbenes and lignans [15]. An association has been suggested between polyphenol intake and reduced mortality and morbidity for chronic diseases such as coronary artery disease, certain cancers, neurodegenerative diseases, such as Parkinson’s and Alzheimer’s disease, as well as T2DM. The beneficial health effects of polyphenols have been associated with their ability to act as effective scavengers of most types of oxidising species, such as reactive oxygen (ROS) and nitrogen species (RNS) and their ability to bind to a variety of proteins, including different enzymes. Their pleiotropic effect is associated with multiple molecular targets, such as the modulation of signalling, energy-sensitive, oxidative stress, intestinal microbiota, inflammation- and apoptosis-related pathways, mitochondrial function or epigenetic modifications [16,17,18]. Despite their benefits, polyphenols may also cause adverse effects, especially in vulnerable populations, such as those with genetic polymorphisms in genes related to the polyphenol metabolic pathways. In general, when consumed as food components, polyphenols usually show low toxicity; however, adverse effects might take place for highly fortified foods or when ingested as supplements, which is associated with the risk of carcinogenicity, genotoxicity, hepatotoxicity, endocrine disruption and drug interactions [19,20,21].

Essential fatty acids (FAs) must be obtained from plant sources in the diet. FAs can be classified into three categories based on the number of double bonds present in side chains: saturated FAs (SFAs, no double bonds), monounsaturated FAs (MUFAs, a single double bond) and polyunsaturated FAs (PUFAs, ≥2 double bonds). FAs can be further classified by their carbon chain length and the position of the first double bond from the terminal methyl group (omega; ω; or n−FAs). Among omega-3 PUFAs, α-linolenic (ALA), eicosapentaenoic (EPA), docosahexaenoic (DHA) and docosapentaenoic (DPA) acids are distinguished. The omega-6 PUFAs group includes linoleic (LA) and arachidonic (AA) acids [22].

PUFAs are precursors of various lipid mediators, including pro- and anti-inflammatory eicosanoids and docosanoids. Prostaglandins, prostacyclins, and leukotrienes derived from PUFAs are involved in inflammatory reactions and immune response. EPA is a substrate for the synthesis of anti-inflammatory eicosanoids. DHA is also a precursor of anti-inflammatory and immunomodulatory docosanoids, such as resolvins and protectins. Considering their biological functions mentioned above, both PUFAs omega-3 and PUFAs omega-6 are considered the key factors in the prevention of some undesirable systemic reactions, such as autoimmune response. Moreover, PUFAs play a significant role in chronic diseases, including cardiovascular disorders, cancers and diabetes mellitus [23,24,25]. Recent studies have also indicated the immunomodulatory nature of omega-3 polyphenols and PUFAs [26,27].

Routine laboratory tests in patients with T2DM include, among others, glucose concentration, glycated haemoglobin percentage (HbA1c), lipid profile, liver enzyme activity, creatinine levels, as well as peripheral blood counts. The last investigation provides valuable information on proven and suggested markers of ongoing inflammation, including leukocyte counts (WBC), platelet counts (PLT), mean platelet volume (MPV), platelet-to-lymphocyte ratio (PLR), neutrophil-to-lymphocyte ratio (NLR) and medium platelet volume-to-lymphocyte ratio (MPVLR) [28,29,30]. Recent data indicate that MPVLR, calculated from blood counts, is associated with a risk of developing chronic micro- and macrovascular diabetic complications [31,32]. The importance of NLR and PLR in the development of microvascular complications [33] as predictive and prognostic markers in the development of diabetic nephropathy and retinopathy has also been emphasised [34,35].

However, there is still a lack of data on diabetics’ adherence to the principles of non-pharmacological diabetes therapy, their use of the individual food ingredients and their impact on cardiometabolic parameters and inflammatory markers. Therefore, in our study, we assessed the relationship between polyphenols and FAs (both total and subclasses) consumption and selected metabolic and inflammatory markers in T2DM patients.

## 2. Materials and Methods

Study population.

The study enrolled 129 patients (49 men, 80 women; mean age 64.1 ± 9.8 years) with T2DM, diagnosed based on the WHO criteria, under the care of the Diabetes Outpatient Clinic of the University Teaching Hospital No. 1 in Łódź. Each patient was educated on a diabetic diet at least three months prior to enrolment to the study, according to actual American Diabetes Association recommendations [36]. The exclusion criteria were as follows: infection, recognised cancer disease, anaemia, coronary artery disease, chronic inflammatory disease, liver disease and pregnancy. The study was performed under the guidelines of the Helsinki Declaration for human research and approved by the Medical University of Lodz Committee on the Ethics of Research in Human Experimentation (approval number RNN/182/17/KE). All enrolled subjects provided written informed consent. All participants underwent a comprehensive physical examination, including blood pressure measurements. Anthropometric measurements, such as body weight, height, waist circumference, were also performed and used as the basis for the calculation of BMI and waist-to-hip ratio (WHR).

For every participant, data were collected related to education level, physical activity, current smoking, duration of diabetes, following a diabetic diet, familial diabetes, diabetes complications and treatment, and a blood sample was taken to determine the blood cell count, haemoglobin, haematocrit, MPV, lipid profile (total cholesterol—TCH, TG, LDL-cholesterol—LDL-CH, HDL-cholesterol—HDL-CH), aspartate and alanine aminotransferase (AST, ALT) activity, serum creatinine concentration, FG and HbA1c. Laboratory parameters were measured using standard diagnostic procedures. The morphology parameters such as platelet, neutrophile, lymphocyte counts and MPV were used as the basis for calculation of the inflammatory rates: PLR, NLR and MPVLR.

Dietary questionnaire and the estimation of FAs intake.

To estimate the nutritional value of a diet, the Food Frequency Questionnaire (FFQ) was used (FFQ is included in the Appendix A). FFQ enabled estimating the amount of individual products in patients’ diets over the last year, and based on these assessments, the dietary intakes (in grams) were calculated. The data were entered into an authorial worksheet to estimate the content of the vegetable phenolic compounds. The worksheet was based on data from Phenol Explorer (www.phenol-explorer.eu; accessed on 30 June 2021), an online comprehensive database on polyphenol contents in foods. The worksheet included the following classes: total quantity of vegetable and fruit phenolic compounds, flavonoids (flavons, flavanols, catechins, procyanidins, anthocyanins, teaflavins, dihydrochalcones, isoflavonoids), flavan-3-ols, phenolic acids (hydroxybenzoic acid, hydroxycinnamic acid), stilbenes and lignans. The data from FFQ were also entered into Aliant software (Cambridge Diagnostics, Warsaw, Poland) to estimate the intake of PUFAs omega-3 (including EPA, DHA), PUFAs omega-6, cholesterol and other food components (SFAs, MUFAs).

Validation of the Food Frequency Questionnaire (FFQ) was carried out by determining its validity and reliability using the test–retest method with an interval of 2 weeks. Ninety three participants aged 21–65 years were enrolled in this process. Completion of the questionnaire was preceded by instructions on how to fill out the form, and the person responsible for the study was present in the room while the participants were completing the FFQ. The reliability of the questionnaire was estimated by determining the Bland–Altman Index and the Spearman rank correlation coefficient. Comparing FFQ1 (test) and FFQ2 (retest), the mean intakes were similar across the analysed nutrients. The Bland–Altman index, the most important FFQ conformance index, was 2.69% for polyphenols and 1.79% for FAs with a desired value of less than 5%. For all studied groups of nutritional compounds, a strong and statistically significant correlation was found (Rs > 0.9, *p* < 0.05). The data from the validation process are shown in the Appendix A).

Statistical analysis.

The median values were used for dividing polyphenol or PUFAs omega 3 high and low dietary intakes. The variables were examined for normality (Shapiro–Wilk test). The χ2 test was used to compare categorical variables; the Mann–Whitney U test was used for continuous variables because these data did not fit a normal distribution (comparison between groups with high and low intake). Correlations of dietary intake food compounds with selected parameters from the metabolic and inflammatory profile were assessed using Spearman’s correlation test. A series of multivariable linear regression models were fit, first unadjusted (Model 1); adjusted for age and gender (Model 2); adjusted for smoking and physical activity (Model 3); and adjusted for BMI and WHR (Model 4). Variables with a skewed distribution (HbA1c) were log-transformed prior to being used in statistical models. *p*-values < 0.05 were considered significant. Statistical analysis was conducted using Statistica v. 13.1 (Statsoft, Krakow, Poland).

## 3. Results

Clinical characteristics of study participants

The demographic and clinical characteristic of all T2DM patients who participated in the study is shown in the Appendix A. In 51 patients, the duration of diabetes exceeded 5 years, and in 52 patients, T2DM was diagnosed in the previous year. The participant group was predominantly female. The fraction of participants with a high educational level and the fraction with a high physical activity level were the smallest. A total of 110 diabetic patients were overweight, including 82 obese patients in this group (BMI > 29.99 kg/m^2^). The demographic characteristics of the groups with low and high polyphenols or PUFAs omega 3 intakes are shown in Table 1. No significant relation between gender, physical activity, following of diabetic diet or familial diabetes and low or high intake of polyphenols or PUFAs omega 3 was observed.

We found that T2DM patients, independently of their declaration of diabetic diet adherence, consumed different amounts of polyphenols and fatty acids, including PUFAs omega 3. Additionally, even though 73% of patients in our study group declared adherence to diabetic-appropriate diet (low calories, low glycaemic index, higher intake of dietary fibre and vegetables), almost 30% of them had a fraction of glycated haemoglobin under 7%. Table 2 illustrates the dietary intake of food components estimated in diabetic patients depending on high or low intake of polyphenols and PUFAs omega 3. High dietary intake was defined as intake equal and higher than the median of the estimated intake of polyphenols or selected classes of polyphenols (flavonoids, flavan-3-ols, stilbenes, phenolic acids, lignans) and omega 3 fatty acids. The median values for key food components are shown in the Appendix A (Appendix A). We observed that in the group with a high intake of polyphenols and SFAs, PUFAs omega 3 and cholesterol were significantly higher. Additionally, as expected, the significantly higher levels of flavonoids, flavan-3-ols, phenolic acids, stilbenes and lignans were in the high polyphenol intake group (Table 2). In the group with a high PUFAs omega 3 intake, the significantly higher level was observed for total polyphenols and its classes (flavonoids, phenolic acids, stilbenes, lignans), SFAs, MUFAs, cholesterol and as expected for total PUFAs, PUFAs omega 6, EPA + DHA (Table 3).

Importantly, the effect of a high polyphenol intake on fasting glycaemia and a high PUFAs omega 3 intake on the HbA1c fraction was reported (Table 3). In the group with high total polyphenol intakes, the concentration of fasting glycaemia was lower but not statistically significant (Table 3). The significant decreasing effect of a high PUFAs omega 3 intake was found for the HbA1c fraction (Table 3). We did not observe significant differences between high and low consumption of polyphenols or PUFAs omega 3 for BMI, WHR, waist circumference values, lipid parameters (TCH, LDL, HDL and TG) and inflammatory biomarkers, such as NLR, PLR and MPVLR (Table 3). Additionally, the consumption of polyphenols or omega 3 did not influence blood pressure, serum creatinine concentration, both alanine transaminase and asparagine transaminase activities or blood cell counts (data not shown; these parameters for all study participants are present in the Appendix A).

The most important findings from our study suggest that the intake of polyphenols may modulate the FG level and weakly affect the HbA1c amount. As it was shown above, total polyphenol consumption influences FG but not significantly. Nevertheless, in the group with high intakes of flavonoids (the main group of polyphenols), the glucose concentration was significantly lower compared to the group characterised by low flavonoid intake (*p* < 0.05, Figure 1). The analysis of the association showed that polyphenols, especially flavonoids, influence the fasting glucose concentration; a negative association was observed between glucose level and total polyphenol, flavonoid, flavan-3-ol and stilbene intake (Table 4). Polyphenols, besides stilbenes, did not modulate the fraction of HbA1c. As it was shown in Table 4, stilbenes can modulate (negative association) the concentration of glucose and also HbA1c; the higher intake of stilbenes decreases both parameters but not significantly (for HbA1c, 6.92 ± 1.42% in low intake vs. 6.68 ± 1.12% in high intake, *p* = 0.351; for FG, 136 ± 42 mg/dL in low intake vs. 130 ± 33 mg/dL in high intake, *p* = 0.414). The observed associations are weak (Rs < 0.3) and not statistically significant but may suggest the trend that consumption of polyphenols impacts glucose concentration and the intake of food rich in stilbenes modulates not only FG but also HbA1c amount in diabetic patients.

Moreover, the intake of omega 3 fatty acids, including both eicosapentaenoic (EPA) and docosahexaenoic (DHA) acids, may influence the concentration of glucose and HbA1c. In the group with high consumption of PUFAs omega 3 acids, the FG concentration did not differ (Table 3), but a significant decrease in the amount of HbA1c was reported (*p* < 0.05, Figure 2). Additionally, the intake of EPA + DHA negatively and significantly correlated with both glucose concentration and HbA1c fraction (Table 4). A very weak and not significant correlation between FG or HbA1c and total PUFAs omega 3 (negative correlation) and PUFAs omega 6/3 proportion (positive correlation) was observed (Table 4).

Further analysis using multiple regression revealed significant associations between EPA + DHA intake and HbA1c in the unadjusted model (model 1) and the models adjusted for age and gender (Model 2) or adjusted for BMI and WHR (Model 4). When adjusted for smoking and physical activity (Model 3), this association was altered (Table 5). Any significant association between EPA + DHA intake and glucose concentration was found neither in the unadjusted model nor in the adjusted models (Table 5).

Furthermore, the high dietary intake of the polyphenols or the polyunsaturated fatty acids was independent of familial diabetes or diabetic diet application.

## 4. Discussion

A healthy lifestyle and lifestyle modification, including maintaining a normal body weight, not smoking, being physically active, adhering to a healthy diet and drinking alcohol in a moderate range, can improve metabolic control of the disease and survival among individuals with diabetes [37,38,39]. In our study, most patients reported dieting, but the majority of them rated their activity as low and reported smoking. It should be emphasised that our study involved patients with a recently diagnosed disease as well as people who had been ill for many years. The study population consisted mainly of women. With regard to our results, Raparelli et al. indicate a high percentage of people with coronary heart disease showing medium–high adherence to the MedDiet. Adults with a low adherence to the MedDiet had a higher prevalence of connective tissue disease, were more likely to be current smokers and were more frequently affected by T2DM as compared with the participants with medium–high adherence. No statistically significant differences were observed between subjects with medium–high adherence and subjects with low adherence to the MedDiet with respect to age, sex, BMI, major cardiovascular comorbidities and lifestyle behaviours, such as physical inactivity and polypharmacy [40].

MedDiet is among the most widely studied dietary patterns, as well as in patients with T2DM. The traditional MedDiet is characterised by the consumption of whole grains, legumes, fruits, vegetables, nuts, fish and olive oil, wine in moderation, and a moderate intake of meat, dairy products, processed food and sweets. This diet is also an important source of vitamins, minerals, antioxidants, mono- and polyunsaturated fatty acids, and fibre—all of which provide a wide range of health benefits [41].

The study by Grosso et al. in the Polish population has shown that total dietary polyphenols and some classes of dietary polyphenols were associated with a lower risk of T2DM. Among the main classes of polyphenols, flavonoids, phenolic acids and stilbenes were independent contributors to this association. Both subclasses of phenolic acids (hydroxybenzoic acids and hydroxycinnamic acids) were associated with a decreased risk of T2DM, whereas among the subclasses of flavonoids, high intake of flavanols, flavanones, flavones and anthocyanins was significantly associated with decreased risk of T2DM [42]. In another study, also in the Polish population, Grosso et al. reported that high intake of polyphenols was inversely associated with metabolic syndrome and some of its components (body mass index—BMI, waist circumference, blood pressure, triglycerides—TG, fasting glucose concentration—FG) [43].

There is a lack of data on the frequency of use of the individual components of a diabetic diet, especially in the group of patients with T2DM in the Polish population. The research conducted so far has mainly concerned the impact of individual food products enriched with various types of polyphenols or dietary supplements that contain them on the selected parameters of cardiometabolic risk. Such studies were also conducted in a group of patients with T2DM and provided unambiguous data on the beneficial effects of these products on the cardiometabolic profile of diabetics, although they do not always confirm their significant effect on IR, glucose concentration, HbA1c percentage, creatinine, lipid profile and other cardiometabolic risk factors, such as blood pressure, anthropometric parameters (body weight, waist circumference, hips, BMI and WHR) and body fat content [44,45,46,47,48,49,50,51,52,53,54]. When analysing these data, it is necessary to take into account the large discrepancy in the amount of food products used, the duration of the studies, as well as the frequent lack of consideration of the diet used by the participants during the study. It should be emphasised that our study provides detailed data on the quantity and quality of selected food ingredients. Additionally, we included type 2 diabetics who were informed about the principles of the diet, regardless of whether or not they followed it. Therefore, it is a representative group of patients with whom we deal with everyday medical practice.

There are a few studies on the consumption of polyphenols contained in all meals, especially in the group of patients with T2DM. One of such few studies is the TOSCA.IT study conducted on a group of 2573 patients with T2DM (1535 men and 1038 women, aged 50 to 75 years). In that study, the mean consumption of polyphenols was estimated at about 700 mg/day, with flavonoids and phenolic acids representing the two main classes of polyphenol intake, accounting for 95% of the total polyphenol intake; the remaining 5% was represented by the others three classes (other polyphenols, stilbenes and lignans). From the analysis of three groups of patients with different consumption levels of polyphenols, adjusted for energy intake (lower polyphenol intake 228.5 ± 50.8 mg/1000 kcal, medium 355.0 ± 32.8 mg/1000 kcal and higher 546.4 ± 150.1 mg/1000 kcal, respectively), participants with a higher polyphenol intake were older and had lower BMI and waist and hip circumferences, which was associated, at least to some extent, with better diet quality. In fact, a lower intake of saturated fatty acids and a higher intake of unsaturated fatty acids and fibre were associated with higher consumption of polyphenols. After adjusting for potential confounders (gender, age, BMI, waist and hip circumferences, smoking), patients with the highest intake of energy-adjusted polyphenols had a more favourable cardiovascular risk factors profile compared to patients with the lowest intake. The findings were very similar when the analysis was conducted separately for flavonoids or phenolic acids—the two main classes of polyphenols consumed in this population [55]. It should be emphasised that more than a two-times lower average intake of polyphenols, which were mainly phenols, as well as flavan-3-ols and phenolic acids, was observed in this study. In our study, the mean consumption of polyphenols was estimated at about 1700 mg/day with flavonoids (about 50%), flvan-3-ols (about 25%) and phenolic acids (about 23%) as the dominant classes of consumed polyphenols. Contrary to the TOSCA.IT study, we have carefully calculated the amount of polyphenols consumed. We observed that polyphenols, especially flavonoids, influence the glucose concentration; a negative association was observed between glucose levels and total polyphenol, flavonoid, flavan-3-ol and stilbene intake. Additionally, in the group with a high flavonoid intake, the glucose concentration was significantly lower compared to the group characterised by a low flavonoid intake. Polyphenols, except for stilbenes, did not modulate the HbA1c fraction. Stilbenes can modulate the actual concentration of glucose as well as HbA1c, and it seems that a higher intake of stilbenes decreases both parameters (negative association). Interestingly, we did not find the correlation between polyphenols intake and blood pressure, anthropometric parameters, liver enzymes activity, lipids or creatinine concentration. As we highlighted, there is a lack of data assessing such a relationship in patients with T2DM. The studies conducted so far in groups of patients without diabetes suggest a beneficial effect of supplementation selected products rich in these compounds on these selected cardiovascular risk markers [56,57,58].

A few available studies indicate a significant relationship between enriching the diet with some polyphenols, including those contained in dietary supplements, and reducing inflammation. This also applies to patients with metabolic syndrome (MetS), including patients with T2DM. The most commonly evaluated markers of inflammation, in this case, were C-reactive protein (CRP), ferritin in plasma, oxygen consumption and chemiluminescence in neutrophils, proinflammatory interleukins (IL1β, IL6, IL18), serum total antioxidant capacity (TAC), tumour necrosis factor alpha (TNFα)-alpha, 8-hydroxy-2′-deoxyguanosine (8-OHdG) and nitrite [49,59,60,61,62,63,64]. In our study, we used commonly known markers from blood counts, such as lymphocytes, neutrophils, platelets, as well as the calculated parameters: PLR, NLR and MPVLR, to assess the immune status. The research conducted to date suggests a link between PLR, NLR and MPVLR and the development of chronic diabetes complications (including foot ulcers, lower extremity arterial disease, retinopathy, peripheral neuropathy and nephropathy) [32,35,65,66,67]. In our study, we found no association between polyphenol consumption and these markers, as confirmed by the low proportion of patients enrolled in the study with chronic complications of diabetes. The lack of association between the quality of the diet used and the markers of inflammation (CRP, fibrinogen, IL6) in patients with diabetes is also described in the paper by Liese et al. [68].

The most important finding from our study is that the intake of omega 3 fatty acids, including EPA and DHA, may modulate the glucose level and the HbA1c fraction in T2DM patients. The higher consumption of PUFAs omega 3 significantly decreased HbA1c. Additionally, the intake of EPA and DHA negatively and significantly correlated with both the fasting glucose concentration and HbA1c. We found no association between PUFAs omega 3 and omega 6 consumption and the assessed inflammatory markers (PLR, NLR, MPVLR), which is consistent with the observations of other researchers [69]. It should be emphasised that most of the studies conducted so far evaluating omega 3 and omega 6 PUFAs supplementation indicate their anti-inflammatory properties [70,71]. Studies on a diet with SFAs restriction and MUFAs enrichment conducted so far indicate a beneficial effect on lipid and carbohydrate metabolism, including IR in patients with T2DM [72,73]. It has also been demonstrated that SFAs consumption can have a significant effect on the development of T2DM [74,75]. The substitution of SFAs with PUFAs in patients with MetS is associated with greater reductions in TG and an improvement in endothelial function than obtained with MUFAs, which is independent of weight loss. These preliminary findings raise the possibility that PUFAs may be the unsaturated fat of choice to reduce cardiometabolic risk in patients with MetS [76]. Unlike polyphenol substitution, enriching the diet with omega 3 and omega 6 PUFAs does have to be associated with an improvement in the metabolic profile of T2DM patients, as well as the risk of developing this disease [77]. A study by Golzari et al. did not confirm the beneficial effect of EPA supplementation on the anthropometric parameters, such as BMI, waist circumference and blood pressure values, as well as FG concentration and HbA1c value [78]. High egg consumption did not have an adverse effect on the lipid profile of people with T2DM in the context of increased MUFAs and PUFAs consumption. This study suggests that a high-egg diet can be included safely as part of the dietary management of T2DM, and it may provide greater satiety [79].

## 5. Conclusions

Our study indicates a low level of adherence to the recommendations for non-pharmacological treatment of T2DM, despite the declaration of using a diabetic diet. Among the assessed food ingredients, we found a positive effect of high flavonoid, PUFAs omega 3 and stilbene intake on the markers of carbohydrate metabolism balance and the absence of such an effect on other cardiometabolic and inflammatory markers.

## Figures and Tables

**Figure 1 nutrients-14-01083-f001:**
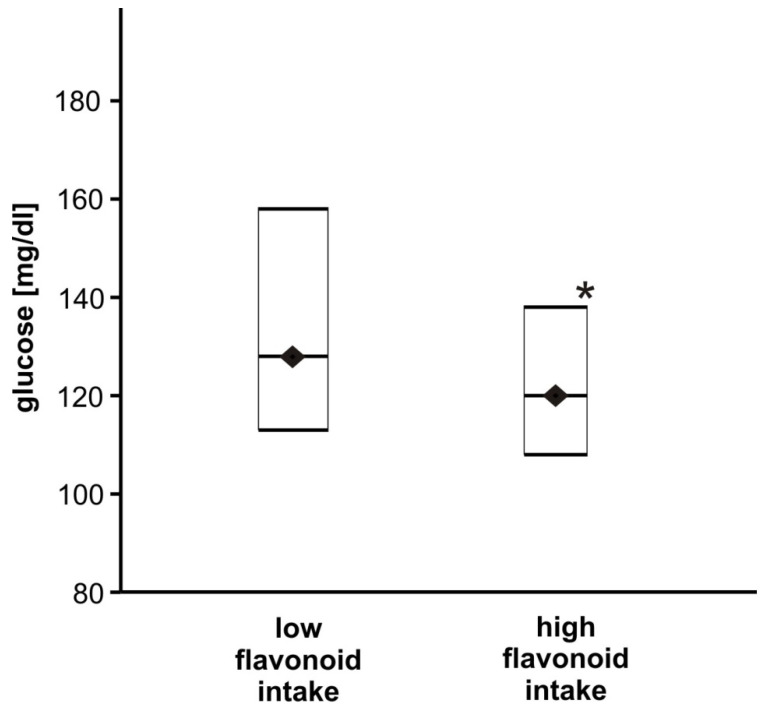
The comparison of glucose concentration between T2DM patients with high and low flavonoid intakes. Data are shown as medians and interquartile ranges (Q1; Q3). The glucose concentration was significantly lower for the high flavonoid intake group (*n* = 65) compared to the group characterised by low flavonoid intake (*n* = 64), * *p* < 0.05.

**Figure 2 nutrients-14-01083-f002:**
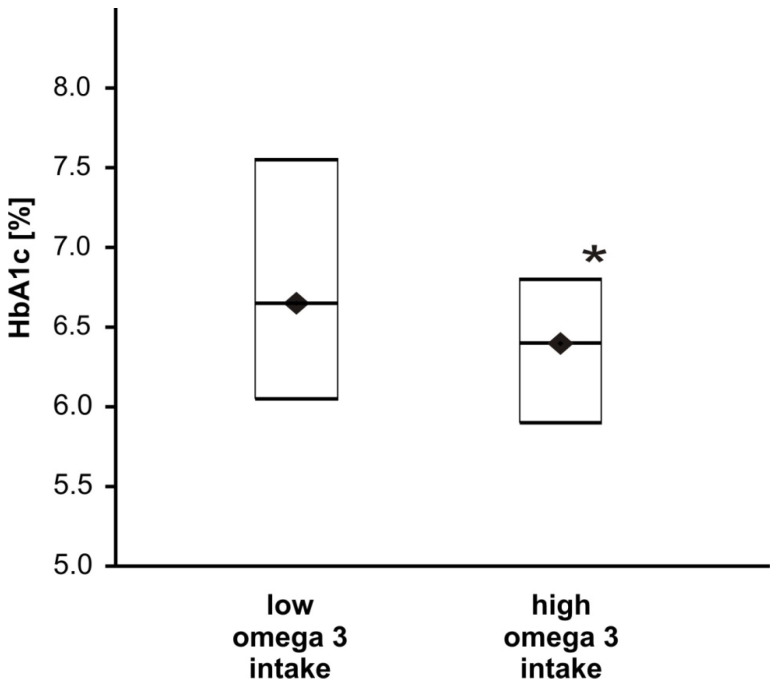
The effect of omega 3 intake on the HbA1c fraction in T2DM patients. Data are shown as medians and interquartile ranges (Q1; Q3). The HbA1c fraction was significantly lower for the group with high omega 3 intake (*n* = 65) compared to the group characterised by low omega 3 intake (*n* = 64), * *p* < 0.05.

**Table 1 nutrients-14-01083-t001:** Demographic characteristics of T2DM patients depending on the intake of polyphenols or PUFAs omega 3.

Demographic Variables	Polyphenol Intake		Omega 3 Intake	
	low (*n* = 64)	high (*n* = 65)	low (*n* = 64)	high (*n* = 65)
gender: female *n* (%) male *n* (%)	36 (56) 28 (44)	44(68) 21 (32)	39 (61) 25 (39)	41 (63) 24 (37)
physical activity: low *n* (%) medium *n* (%) high *n* (%)	39 (61) 20 (31) 5 (8)	40 (62) 19 (29) 6 (9)	41 (64) 15 (23) 8 (13)	38 (58) 24 (37) 3 (5)
duration of diabetes (years]	5 (0; 8)	3 (0; 10)	5 (0; 9)	3 (0;9)
following of diabetic diet *n* (%)	47 (73)	47 (72)	45 (70)	49 (75)
familial diabetes *n* (%)	25 (39)	35 (54)	24 (37)	30 (46)

Duration of diabetes is presented as median and interquartile ranges (Q1; Q3). The percentages (%) were calculated as the fraction of low or high intake.

**Table 2 nutrients-14-01083-t002:** Descriptive characteristic of dietary consumption of key food components in the T2DM patient groups with low and high polyphenol or PUFAs omega 3 intakes.

	Polyphenol Intake		PUFAs Omega 3 Intake	
	low	high	low	high
polyphenols (mg/day)	958 (503; 1410)	2076 (1845; 2672)	1410 (585; 1878)	1817 (1424; 2449) *
flavonoids (mg/day)	406 (179; 694)	1101 (946; 1301) *	773 (285; 1021)	936 (595; 1220) *
flavan-3-ols (mg/day)	225 (64; 390)	648 (524; 783) *	442 (102; 622)	432 (270; 670)
phenolic acids (mg/day)	235 (132; 353)	452 (285; 704) *	244 (147; 357)	413 (280; 745) *
stilbenes (mg/day)	0.1 (0.02; 0.52)	0.54 (0.07; 1.05) *	0.08 (0.02; 0.62)	0.41 (0.09; 0.80) *
lignans (mg/day)	11.9 (6.9; 23.5)	23.9 (15.4; 44.7) *	9.7 (6.5; 21.4)	25.9 (16.4; 46.2) *
SFAs (g/day)	24.0 (13.7; 30.4)	28.2 (22.3; 38.3) *	20.8 (12.9; 28.9)	29.4 (24.3; 40.3) *
MUFAs (g/day)	33.5 (17.9; 45.7)	38.3 (22.4; 52.1)	22.4 (16.9; 36.4)	44.4 (36.4; 61.4) *
PUFAs (g/day)	14.1 (7.9; 23.9)	18.4 (10.6; 27.4)	9.3 (5.9; 14.2)	24.5 (18.6; 31.9) *
PUFAs omega 3 (g/day)	1.87 (1.18; 3.42)	2.98 (1.90; 4.74) *	1.37 (1.03; 1.87)	3.94 (3.08; 6.51)
PUFAs omega 6 (g/day)	11.5 (6.6; 17.7)	14.8 (8.9; 21.3)	7.6 (4.4; 12.3)	18.9 (14.3; 23.7) *
EPA + DHA (g/day)	0.29 (0.17; 0.65)	0.37 (0.23; 0.83)	0.23 (0.11; 0.35)	0.65 (0.29; 1.32) *
cholesterol (mg/day)	221 (127; 313)	310 (207; 468) *	196 (126; 311)	311 (248; 440) *

Data are shown as median (MD) and interquartile range (Q1; Q3). Significant differences were observed between high and low intake, * *p* < 0.05 or less.

**Table 3 nutrients-14-01083-t003:** Biochemical, inflammatory and anthropometric parameters in the T2DM patient groups with low and high polyphenol or PUFAs omega 3 intakes.

	Polyphenols		PUFAs Omega 3	
	low	high	low	high
FG (mg/dL)	127 (113; 155)	119 (108; 138)	122 (112; 157)	121 (108; 138)
HbA1c (%)	6.5 (5.9; 7.3)	6.4 (6.0; 6.9)	6.6 (6.0; 7.5)	6.4 (5.9; 6.8) *
TCH (mg/dL)	180 (144; 207)	185 (149; 213)	185 (152; 216)	177 (145; 201)
LDL-TCH (mg/dL)	100 (73; 132)	103 (79; 124)	101 (75; 125)	102 (76; 131)
HDL-CH (mg/dL)	47 (41; 55)	49 (40; 61)	49 (40; 55)	49 (40; 61)
TG (mg/dL)	141 (104; 201)	138 (103; 178)	138 (108; 199)	139 (101; 174)
PLR	114 (91; 139)	106 (85; 142)	107 (82; 131)	119 (89; 142)
NLR	2.2 (1.6; 2.8)	2.2 (1.6; 2.9)	2.2 (1.5; 2.9)	2.2 (1.6; 2.9)
MPVLR	5.0 (4.1; 6.2)	4.6 (4.0; 5.9)	5.2 (4.0; 6.0)	4.8 (4.1; 6.3)
BMI (kg/m^2^)	31.2 (28.3; 35.2)	34.0 (27.1; 38.1)	33.0 (28.1; 35.9)	32.0 (28.2; 36.9)
waist circumference (cm)	108 (102; 115)	111 (101; 119)	110 (101; 118)	108 (102; 117)
WHR	0.96 (0.91; 1.01)	0.97 (0.92; 1.02)	0.97 (0.91; 1.02)	0.96 (0.92;1.02)

Data are shown as median (MD) and interquartile range (Q1; Q3). The significance level of HbA1c was observed for high PUFAs omega 3 intakes compared to PUFAs omega 3 low intakes, * *p* < 0.05. FG- fasting glucose concentration; HbA1c—glycated haemoglobin; TCH—total cholesterol; HDL-CH—HDL-cholesterol; LDL-CH—LDL-cholesterol; TG—triglycerides; MPVLR—mean platelet volume-to-lymphocyte ratio; NLR—neutrophil-to-lymphocyte ratio; PLR—platelet-to-lymphocyte ratio; BMI—body mass index; WHR—waist-to-hip ratio.

**Table 4 nutrients-14-01083-t004:** Correlation coefficients (Rs) for HbA1c or glucose concentration and food components in T2DM patients.

	HbA1c	Glycaemia
Total polyphenols	−0.045993	−0.139093
Flavonoids	−0.078909	−0.143564
Flawan-3-ols	−0.010229	−0.110597
Phenolic acids	−0.075401	−0.105316
Stilbenes	−0.149940	−0.108842
Lignans	−0.076959	−0.054942
MUFAs	−0.065301	0.016014
Total PUFAs	−0.082863	−0.075725
PUFAs omega 3	−0.130564	−0.111073
PUFAs omega 6	−0.075280	−0.070239
EPA + DHA	−0.188344 *	−0.245573 *
Omega 6/3 proportion	0.121846	0.138595
Cholesterol	−0.024314	0.068974

DHA—docosahexaenoic acid; EPA—eicosapentaenoic acid; MUFAs—monounsaturated fatty acids; PUFAs—polyunsaturated fatty acids. The association between the variables was estimated by the simple Spearman rank test; * *p* < 0.05.

**Table 5 nutrients-14-01083-t005:** Association of EPA + DHA intake with HbA1c and FG.

	Beta	SE	*p*
HbA1c (%) Model 1 Model 2 Model 3 Model 4	0.000580 0.000569 0.000466 0.000586	0.000254 0.000026 0.000241 0.000257	0.024 * 0.028 * 0.055 0.024 *
FG (mg/mL) Model 1 Model 2 Model 3 Model 4	−0.000038 −0.000039 −0.000026 −0.000041	0.000032 0.000032 0.000031 0.000032	0.243 0.224 0.399 0.205

Model 1: unadjusted. Model 2: adjusted for age and gender. Model 3: adjusted for smoking and physical activity. Model 4: adjusted for BMI and WHR. The significant association was observed between intake of EPA + DHA and HbA1c, * *p* < 0.05

## Data Availability

The data presented in this study are available on request from the corresponding author. The data are not publicly available due to ethical reasons.

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
