# Peer review of "Dietary Intake of Polyphenols or Polyunsaturated Fatty Acids and Its Relationship with Metabolic and Inflammatory State in Patients with Type 2 Diabetes Mellitus"

_nutrients, 2022, doi:10.3390/nu14051083_

Round 1

Reviewer 1 Report

The authors attempt to describe the association between dietary polyphenols and polyunsaturated fatty acid (PUFA), saturated fatty acid (SFA) to the metabolic and inflammatory markers in Polish population with type 2 diabetes mellitus (T2DM). The positive feature of the paper is being the first to report association between dietary polyphenols and PUFA to fasting glucose concentration (FG) and glycated hemoglobin percentage (HbA1c) in Polish T2DM patients. The lack of analyzed data visualization, focus on the aim of the study, and details of how multilinear regression is performed are the major drawbacks to the paper. The authors should consider putting descriptive details on characteristics that do not offer further explanation on the data presented in the result section in the supplementary material section. The authors should also consider presenting their data more clearly and focusing their discussion on their data. In addition, putting more details on how the authors established each model in their multilinear regression analyses will be very helpful. Overall, the manuscript offers an interesting perspective on the effect of dietary polyphenols or PUFAs in the glucose concentration or glycated hemoglobin percentage in T2DM patients.

Major points:

  1. In the Results section, p.5, Table 1, the authors might want to consider trimming the details presented in Table 1. The authors can put the minute details of the patients in the supplementary material section. Table 1 will be more informative if the authors break down the details of the demographics based on the stratification on high and low flavonoid intake groups as well as high and low PUFA intake groups.
  2. In Results section, p. 6, first paragraph, the authors state that there are significant association between polyphenol and FG or omega 3 and HbA1c. The authors should consider referring to the figures that support the statement.
  3. In Results section, p. 6 and 7, Tables 2 and 3, the authors should consider stratifying the groups, including p-values of their statistical analyses in this table or the authors might want to consider showing the data in separate scattered boxplots with p-values after stratification. In addition, the authors might want to consider trimming the other parameters presented in the table.
  4. In Results section, p. 7, first paragraph, the authors state that the intake of polyphenols and omega 3 fatty acids affect the fasting glucose level and HbA1c. In this paper, the authors show separately that high polyphenols intake is associated with lower FG and high PUFA intake is associated with lower HbA1c. It is concerning that te authors have not shown having high intake of both polyphenols and PUFAs is associated with improved FG and HbA1c as stated as the aim of the study. Another statement in this paragraph is glucose level is negatively associated with total polyphenol, flavonoid, flavan-3-ol, and stilbene intake. This reviewer agrees that the negative trend is observed, however, it is not statistically significant. Will this translate into any significant effect?
  5. In Results section, p. 8, Table 4, the authors should consider stratifying the groups when presenting the data.
  6. In Results section, p.8, line 279, the authors state that “Polyphenols, besides stilbenes, did not modulate the fraction of HbA1c”. A citation is needed for this statement. If this statement is from the study, the authors should refer it to the table or figure that supports the statement.
  7. In Results section, p. 8, lines 279-281, there is a negative trend observed in the association between stilbenes and fasting glucose concentration or HbA1c, however, it is not statistically significant. Are there other measured parameters that the authors use to determine the causal effect of stilbenes on fasting glucose concentration and HbA1c?
  8. In Results section, p. 9, second paragraph, the authors discuss about the multilinear regression models and analyses on the EPA+DHA intake and HbA1c. It is unclear how the authors come up with each model. With each choice of covariates, which covariates are influencing the outcome of the association, and which are confounding? The authors should consider doing stepwise regression when performing multilinear regression analyses.
  9. In Discussion section in general, the authors should consider discussing their data and show how their data are supported by reports in the literature.
  10. In Discussion section, p. 10, lines 313-315, the authors emphasize that their patients are a mix of newly diagnosed T2DM patients and longtime T2DM patients. How will the authors expect this factor to affect the results of the study?
  11. In Discussion section, p. 10, first paragraph, the authors mention that benefit of adherence to Mediterranean diet. Do the patients in the current study adhere to similar diet? What kind of diets do the patients in this study adhere to?
  12. In Discussion section, p. 11, second paragraph, the authors discuss the literature reports on association between polyphenols and inflammatory markers. It is unclear the proportion of the patients who adhere to high polyphenol intake compared to low polyphenol intake. Do you have enough number of patients in both groups?

Minor points

  1. In Abstract section, p.1, line 17, the authors should define “PUFA” because it is the first time the word “PUFA” appears in the text.
  2. In Abstract section, p.1, line 18, the authors should define “T2DM” because it is the first time the word “T2DM” appears in the text.
  3. In Abstract section, p.1, line 21, the authors should define “HbA1c” because it is the first time the word “HbA1c” appears in the text.
  4. In Abstract section, p.1, lines 25-26, the authors should define “EPA” and “DHA” because it is the first time the words “EPA” and “DHA” appear in the text.
  5. In Abstract section, p.1, line 27, the authors should define “BMI” and “WHR” because it is the first time the words “BMI” and “WHR” appear in the text.
  6. In Introduction section, p.2, line 95, the authors refer to reactive nitrogen species as “RONS”, it should be “RNS”. RONS encompasses both reactive oxygen and nitrogen species.
  7. In Introduction section, p.3, lines 111 and 113, the authors should replace the “type 2 diabetes” with T2DM because the word “type 2 diabetes” has been defined as T2DM earlier in the text.
  8. In Introduction section, p.3, line 122, the authors might want to consider changing the sentence “… double bond on methyl terminal…” to “double bond from the terminal methyl group..”
  9. In Introduction section, p.4, line 165, there is a mistype on “Fas” when referring to “FAs”.
  10. In the Results section, p. 5, Table 1, the unit for the “Duration of diabetes” is missing.
  11. In the Results section, p. 6, Table 2, the rows are misaligned with the values of each parameter. There is a mistype of “Fas” when referring to “FAs”. In addition, the abbreviations of the acronyms are not necessary because they are defined in the text.
  12. In the Results section, p. 7, Table 3, the rows are misaligned with the values of each parameter. There is a mistype of “leucocytes” when referring to “leukocytes”. In addition, the abbreviations of the acronyms are not necessary because they are defined in the text.
  13. In the Results section, p.7 and 8, Table 4, please ensure the title of the table is on the same page as the table. There is also a mistype of “Fas” in the table when referring to “FAs”.
  14. In the Results section, p. 5, lines 216- 129, the authors should use “among” instead of “between”. In addition, the authors might want to consider omitting the information on the 563 children who were excluded in the meta-analyses in the current study.
  15. In the Results section, p.8, line 277, on the first sentence in the paragraph, “… with high flavonoid intake the glucose concentration…”, the authors should put a comma after the word “intake”
  16. In the Discussion section, p. 10, line 342, the authors should consider using other word other than “probant” when referring to the study participants.
  17. In the Discussion section, p. 11, line 369, the authors should put a comma after the word “intake”.

Author Response

Response to Reviewer 1 Comments

Thank the Reviewer’s very much for your time and valuable comments on our manuscript. The responses for all points are below. The changes were introduced into the text of manuscript, as suggested by the Reviewer.

Major points:

Point 1: In the Results section, p.5, Table 1, the authors might want to consider trimming the details presented in Table 1. The authors can put the minute details of the patients in the supplementary material section. Table 1 will be more informative if the authors break down the details of the demographics based on the stratification on high and low flavonoid intake groups as well as high and low PUFA intake groups.

Response 1: Thank the Reviewer’s suggestion. Table 1 was changed in the amended version of manuscript, and now presents the T2DM patient characteristics based on the stratification on high and low polyphenol intake groups as well as high and low PUFA omega 3 intake groups. The older version of Table 1 with demographic and clinical characteristics of all participants was included into Supplementary as Table S4.

Point 2: In Results section, p. 6, first paragraph, the authors state that there are significant association between polyphenol and FG or omega 3 and HbA1c. The authors should consider referring to the figures that support the statement.

Response 2: The Reviewer’s suggestion is very relevant. The description of the results from the study was change in the Results section in the amended version of manuscript: the important observation for glucose/ HbA1c and polyphenols /omega 3 was separated and the referring to the relevant figures or tables was included.

Point 3: In Results section, p. 6 and 7, Tables 2 and 3, the authors should consider stratifying the groups, including p-values of their statistical analyses in this table or the authors might want to consider showing the data in separate scattered boxplots with p-values after stratification. In addition, the authors might want to consider trimming the other parameters presented in the table.

Response 3: According to the Reviewer’s suggestion Tables 2 and 3 were changed in the amended version of manuscript, and now present dietary intake of key food components (Table 2) or biochemical, inflammatory and anthropometric characteristics (Table 3) based on the stratification on high and low polyphenol intake groups as well as high and low PUFA omega 3 intake groups. The data of hematological characteristics of patients were removed from the new version of Table 3. The older version of Table 2 and Table 3 with characteristics of all participants was included into Supplementary as Table S5 and Table S6, respectively.

Point 4: In Results section, p. 7, first paragraph, the authors state that the intake of polyphenols and omega 3 fatty acids affect the fasting glucose level and HbA1c. In this paper, the authors show separately that high polyphenols intake is associated with lower FG and high PUFA intake is associated with lower HbA1c. It is concerning that te authors have not shown having high intake of both polyphenols and PUFAs is associated with improved FG and HbA1c as stated as the aim of the study. Another statement in this paragraph is glucose level is negatively associated with total polyphenol, flavonoid, flavan-3-ol, and stilbene intake. This reviewer agrees that the negative trend is observed, however, it is not statistically significant. Will this translate into any significant effect?

Response 4: Thank the Reviewer’s comment. As it was mentioned above, the Results section in the amended version of manuscript was changed and the description of data for polyphenols/glucose and omega 3/HbA1c was separated as far as possible.

The associations shown in Table 4 for polyphenols, its classes and glucose level are weak and no statistically significant. Therefore, to the better explanation the data the appropriate comment was included in the text in the amended version of manuscript (p. 10):

“The observed associations are weak (Rs<0.3) and not statistically significant, but may suggest the trend that consumption of polyphenols impacts on glucose concentration and the intake of food rich in stilbenes modulates not only FG but also HbA1c amount in diabetic patients.”

Point 5: In Results section, p. 8, Table 4, the authors should consider stratifying the groups when presenting the data.

Response 5: Thank the Reviewer’s comment. To analyse of association between polyphenol or omega 3 intake and FG or HbA1c we used data from all participants. In our opinion this strategy is better than analysis data from stratified groups because of more numerous in groups (n=129 for group of all participants, n=64 for low intake or n=65 for high intake) and variety of data.

Point 6: In Results section, p.8, line 279, the authors state that “Polyphenols, besides stilbenes, did not modulate the fraction of HbA1c”. A citation is needed for this statement. If this statement is from the study, the authors should refer it to the table or figure that supports the statement.

Point 7: In Results section, p. 8, lines 279-281, there is a negative trend observed in the association between stilbenes and fasting glucose concentration or HbA1c, however, it is not statistically significant. Are there other measured parameters that the authors use to determine the causal effect of stilbenes on fasting glucose concentration and HbA1c?

Response 6 and 7: Thank the Reviewer’s comments. The statement that “Polyphenols, besides stilbenes, did not modulate the fraction of HbA1c” was from our study. To better explanation of the statement the appropriate data was included into the text in the amended version of the manuscript (p.10):

 “As it was shown in Table 4, stilbenes can modulate (negative association) the concentration of glucose and also HbA1c; the higher intake of stilbenes decreasing both parameters but not significantly (for HbA1c 6.92 ± 1.42% in low intake vs. 6.68 ± 1.12% in high intake, p=0.351; for FG 136 ± 42 mg/dl in low intake vs. 130 ± 33 mg/dl in high intake, p=0.414).

Point 8: In Results section, p. 9, second paragraph, the authors discuss about the multilinear regression models and analyses on the EPA+DHA intake and HbA1c. It is unclear how the authors come up with each model. With each choice of covariates, which covariates are influencing the outcome of the association, and which are confounding? The authors should consider doing stepwise regression when performing multilinear regression analyses.

Response 8: Thank the Reviewer’s comments. The multivariable linear regression models used in this study is the supplementation of the simple correlation analysis, when the EPA + DHA consumption is significantly associated with glucose concentration and HbA1c fraction. On the one hand, the choice of covariates included in multivariable regression models such as age, gender (model 2), smoking and physical activity (model 3), BMI and WHR (model 4) was based on variables related to healthy lifestyle and non-modifiable variables such as age and gender which can be associated with cardiovascular complication in diabetes. Also, we done the stepwise regression to construct the regression models. There were any significant association between EPA + DHA intake and age, BMI or WHR; or between HbA1c fraction and age, BMI or WHR; or between BMI and WHR; or between glucose concentration and age, BMI or WHR. Therefore, the covariates used in our models are rather influencing the outcome of the association, than confounding covariates.

Point 9: In Discussion section in general, the authors should consider discussing their data and show how their data are supported by reports in the literature.

Response 9: The text of the discussion has been changed as recommended

Point 10: In Discussion section, p. 10, lines 313-315, the authors emphasize that their patients are a mix of newly diagnosed T2DM patients and longtime T2DM patients. How will the authors expect this factor to affect the results of the study?

Response 10: Thank you very much for this comment. Each patients was educated on a diabetic diet at least three months prior to enrollment to the study, according to actual American Diabetes Association recommendation (text added to the manuscript). It is difficult to predict whether dietary adherence worsens with the duration of diabetes. It all depends on the patient's individual approach. We enrolled 129 consecutive patients for the study, so we have a group that is representative of the population.

Point 11: In Discussion section, p. 10, first paragraph, the authors mention that benefit of adherence to Mediterranean diet. Do the patients in the current study adhere to similar diet? What kind of diets do the patients in this study adhere to?

Response 11: The patients in this study were previously informed about the principles of diabetes diet based mainly on Mediterranean diet combined with a reduction in calories, if necessary, according to ADA recommendation. They should follow such a diet. For various reasons (financial issues, eating habits, etc ...), they were not always able to fully implement it.

Point 12: In Discussion section, p. 11, second paragraph, the authors discuss the literature reports on association between polyphenols and inflammatory markers. It is unclear the proportion of the patients who adhere to high polyphenol intake compared to low polyphenol intake. Do you have enough number of patients in both groups?

Response 12: We added to the amended version of manuscrip Table 1, in which we included number of patinets from high or low polyphenol intake group. We believe this is a large enough group of patients from which to draw conclusions.

Minor points:

Point 1: In Abstract section, p.1, line 17, the authors should define “PUFA” because it is the first time the word “PUFA” appears in the text.

Response 1: We have added a deffinition of “PUFA” in abstract. Our previous decision was dictated by verbal limits.

Point 2: In Abstract section, p.1, line 18, the authors should define “T2DM” because it is the first time the word “T2DM” appears in the text.

Response 2: We have added a deffinition of “T2DM” in abstract. Our previous decision was dictated by verbal limits.

Point 3: In Abstract section, p.1, line 21, the authors should define “HbA1c” because it is the first time the word “HbA1c” appears in the text.

Response 3: We have added a deffinition of “HbA1c” in abstract. Our previous decision was dictated by verbal limits.

Point 4: In Abstract section, p.1, lines 25-26, the authors should define “EPA” and “DHA” because it is the first time the words “EPA” and “DHA” appear in the text.

Response 4: We have added a deffinition of “EPA” and “DHA” in abstract. Our previous decision was dictated by verbal limits.

Point 5: In Abstract section, p.1, line 27, the authors should define “BMI” and “WHR” because it is the first time the words “BMI” and “WHR” appear in the text.

Response 5: We have added a deffinition of “BMI” and “WHR in abstract. Our previous decision was dictated by verbal limits.

Point 6: In Introduction section, p.2, line 95, the authors refer to reactive nitrogen species as “RONS”, it should be “RNS”. RONS encompasses both reactive oxygen and nitrogen species.

Response 6: Thank the Reviewer’s comment. We corrected these mistake in the manuscript.

Point 7: In Introduction section, p.3, lines 111 and 113, the authors should replace the “type 2 diabetes” with T2DM because the word “type 2 diabetes” has been defined as T2DM earlier in the text.

Response 7: The abbreviation is provided in the manuscript.

Point 8: In Introduction section, p.3, line 122, the authors might want to consider changing the sentence “… double bond on methyl terminal…” to “double bond from the terminal methyl group..”

Response 8: Corrected in the text according to the note.

Point 9: In Introduction section, p.4, line 165, there is a mistype on “Fas” when referring to “FAs”.

Response 9: Corrected in the text according to the note.

Point 10: In the Results section, p. 5, Table 1, the unit for the “Duration of diabetes” is missing.

Response 10: Thank the Reviewer’s comment. In the amended version of manuscript is a new version of Table 1.

Point 11: In the Results section, p. 6, Table 2, the rows are misaligned with the values of each parameter. There is a mistype of “Fas” when referring to “FAs”. In addition, the abbreviations of the acronyms are not necessary because they are defined in the text.

Response 11: Thank the Reviewer’s comment. In the amended version of manuscript is a new version of Table 2. According to Reviewer’s suggestion the abbreviations of the acronyms were removed.

Point 12: In the Results section, p. 7, Table 3, the rows are misaligned with the values of each parameter. There is a mistype of “leucocytes” when referring to “leukocytes”. In addition, the abbreviations of the acronyms are not necessary because they are defined in the text.

Response 12: Thank the Reviewer’s comment. In the amended version of manuscript is a new version of Table 3. According to Reviewer’s suggestion the abbreviations of the acronyms were removed.

Point 13: In the Results section, p.7 and 8, Table 4, please ensure the title of the table is on the same page as the table. There is also a mistype of “Fas” in the table when referring to “FAs”.

Response 13: Thank the Reviewer’s comment. In the amended version of manuscript Table 4 was corrected.

Point 14: In the Results section, p. 5, lines 216- 129, the authors should use “among” instead of “between”. In addition, the authors might want to consider omitting the information on the 563 children who were excluded in the meta-analyses in the current study.

Response 14: I don’t understand the Reviewer’s comment.

Point 15: In the Results section, p.8, line 277, on the first sentence in the paragraph, “… with high flavonoid intake the glucose concentration…”, the authors should put a comma after the word “intake”

Response 15: Thank the Reviewer’s comment. In the amended version of manuscript it was corrected.

Point 16: In the Discussion section, p. 10, line 342, the authors should consider using other word other than “probant” when referring to the study participants.

Response 16: Corrected in the text according to the note.

Point 17: In the Discussion section, p. 11, line 369, the authors should put a comma after the word “intake”.

Response 17: Corrected in the text according to the note.

Reviewer 2 Report

Manuscript ID: nutrients-1552901

The aim of this paper was to better understand the relationship between dietary polyphenols and/or PUFAs with inflammatory and metabolic markers in 129 type 2 diabetic patients. The authors stated that the most important findings of the study were that the intake of n3 fatty acids modulate glucose level and HbA1c in these patients. Moreover, a positive effect of flavonoids and stilbenes consumption was also present.

Although the topic is very interesting in the current panorama, the manuscript presents some areas of weakness and the results of the work are not particularly innovative. There are many papers showing positive effects of polyphenols and omega 3 PUFA on carbohydrate metabolism in diabetic patients (just to name a few: Dardashti Pour E. et al Clinical Nutrition ESPEN  2021; 45:134-140. - Sequeira IR et al Nutrients. 2017;9(7):788. - Liu K, et al Am J Clin Nutr. 2018 ;108(2):256-265. - Thota RN, et al. Lipids Health Dis. 2019 26;18(1):31. - Cao H, et al Crit Rev Food Sci Nutr. 2019;59(20):3371-3379).

Furthermore, the work is not well structured:

in the abstract, the AIM of the work is absolutely not mentioned.  

The introduction is too long, particularly evident in the description of the polyphenols and omega 3 (lines 52-91; lines 117-150). It seems more like a review than an article: it must be definitely reduced also clarifying in the introduction what the purpose of the work is.

In the results section (Fig 1 and Fig 2) the authors should clarify which is the cut point for the two different groups (low and high intake of flavonoids, and omega 3, respectively). Furthermore, the authors must necessarily indicate the number of groups that are compared (both in fig 1 and 2).

In the discussion, authors must emphasize, and critically discuss the results they find particularly important. On the contrary, the authors talk about their "most important finding" almost at the end of the discussion (392-396).

Other comments:

  • As is evident from table 4, the correlations are significant only for EPA + DHA. Therefore the authors must emphasize in the text that the results for polyphenols are not significant (pag 7 lines 267-271; pag 8 280-281, pag 12 lines 421-422). Significance is present only for omega 3 PUFAs and this result (although already known) could be more widely discussed.
  • In the text, statements are often made without indicating the relative references. Just for example:
    • pag 2 lines 74-78; line 96 (…including different enzymes) ;  line 99 (…or epigenetic modifications)
    • pag 3 lines 135 -136
    • pag 11 lines 374-376

  • the sentence on page 3 (lines 106-115) would be more appropriate for discussion
  • It would be useful to provide (as Supplemental Material) the food frequency questionnaire, used to estimate the nutritional intake in T2D patients. This would allow to the reader to give a correct evaluation of the questionnaire.
  • please check the use of acronyms in the text, and use them where necessary in place of “Fatty acids” and “type 2 diabetes”
  • Please, proofread and fix some typos, and grammatical mistakes (such as “veetables” Pag 6 line 246; or  “at around” pag 10 line347
  • Table 2 is not well aligned

Author Response

Response to Reviewer 2 Comments

Thank the Reviewer’s very much for your time and valuable comments on our manuscript. The responses for all points are below. The changes were introduced into the text of manuscript, as suggested by the Reviewer.

Point 1: The aim of this paper was to better understand the relationship between dietary polyphenols and/or PUFAs with inflammatory and metabolic markers in 129 type 2 diabetic patients. The authors stated that the most important findings of the study were that the intake of n3 fatty acids modulate glucose level and HbA1c in these patients. Moreover, a positive effect of flavonoids and stilbenes consumption was also present.

Although the topic is very interesting in the current panorama, the manuscript presents some areas of weakness and the results of the work are not particularly innovative. There are many papers showing positive effects of polyphenols and omega 3 PUFA on carbohydrate metabolism in diabetic patients (just to name a few: Dardashti Pour E. et al Clinical Nutrition ESPEN 2021; 45:134-140. - Sequeira IR et al Nutrients. 2017;9(7):788. - Liu K, et al Am J Clin Nutr. 2018 ;108(2):256-265. - Thota RN, et al. Lipids Health Dis. 2019 26;18(1):31. - Cao H, et al Crit Rev Food Sci Nutr. 2019;59(20):3371-3379).

Response 2: Thank the Reviewer’s suggestion and comments. The responses for all points are below. The changes were introduced into the text of manuscript, as suggested by the Reviewer.

We agree that there are many papers showing positive effects of polyphenols and omega 3 PUFA on carbohydrate metabolism in diabetic patients, but in our study we have focused on consumption of polyphenols and fatty acids and its main classes with dietary diet. Also, we have performed the study in the group of T2DM patients from polish population. The sources of polyphenols and omega 3 differ between people from different countries and we hope that our study may contribute to better education of diabetic patients about diet in Poland.

Point 2: Furthermore, the work is not well structured:

In the abstract, the AIM of the work is absolutely not mentioned. The introduction is too long, particularly evident in the description of the polyphenols and omega 3 (lines 52-91; lines 117-150). It seems more like a review than an article: it must be definitely reduced also clarifying in the introduction what the purpose of the work is.

Response 2: The purpose of the work in the abstract has been corrected The introduction was significantly shortened on the basis of the modification of two fragments marked in the review. First of all, too extensive descriptions of the division and content of omega acids and polyphenols were removed

Point 3: In the results section (Fig 1 and Fig 2) the authors should clarify which is the cut point for the two different groups (low and high intake of flavonoids, and omega 3, respectively). Furthermore, the authors must necessarily indicate the number of groups that are compared (both in fig 1 and 2).

Response 3: Thank the Reviewer’s comment. In the amended version of manuscript the number of groups was included into legends of figures (Fig. 1 nad Fig.2). Additionally, the detail description of the group stratification on high and low intake is included into Results section (pp. 7):

“High dietary intake was defined as the intake equal and higher than the median of the estimated intake of polyphenols or selected classes of polyphenols (flavonoids, flavan-3-ols, stilbenes, phenolic acids, lignans) and omega 3 fatty acids. The median values for key food components are shown in Table S5 (Supplementary).”

Point 4: In the discussion, authors must emphasize, and critically discuss the results they find particularly important. On the contrary, the authors talk about their "most important finding" almost at the end of the discussion (392-396).

Response 4: The discussion was modified in line with the comments.

Other comments:

Point 5: As is evident from table 4, the correlations are significant only for EPA + DHA. Therefore the authors must emphasize in the text that the results for polyphenols are not significant (pag 7 lines 267-271; pag 8 280-281, pag 12 lines 421-422). Significance is present only for omega 3 PUFAs and this result (although already known) could be more widely discussed.

Response 5: Thank the Reviewer’s comment. The Results section in the amended version of manuscript was changed and the description of data for polyphenols/glucose and omega 3/HbA1c was separated as far as possible. Also, the presentation of significant and no significant results was corrected.

Point 6: In the text, statements are often made without indicating the relative references. Just for example:

o pag 2 lines 74-78; line 96 (…including different enzymes) ; line 99 (…or epigenetic modifications)

o pag 3 lines 135 -136

o pag 11 lines 374-376

Response 6: We have added the following references as appropriate:

Lorenzo, C.D.; Colombo, F.; Simone, B.; Stockley, C.; Restani, P. Polyphenols and human health. The role of bioavailabity. Nutrients 2021, 13, 273;

Wu, M.; Luo, Q.; Nie, R.; Yang, X.; Tang, Z.; Chen, H. Potenital implications of pholyphenols on aging considering oxidative stress, inflammation, autophagy, and gut microbiota. Crit Rev Food Sci Nutr 2021, 61, 2175-2193

Yahfoufi, N.; Alsadi, N.; Jambi, M; Matar, C. The immunomodulatory and anti-inflammatory role of polyphenols. Nutrients 2018, 10, 1618

Selvakumar, P.; Badgeley, a.; Murphy, P.; Anwar, H.; Sharma,  U.; Lawrence, K.; Lakshmikuttyamma, A. Flavonoids and other polyphenols act as epigenetic modifiers in breast cancers. Nutrients 2020, 12, 761

o pag 3 lines 135 -136

This part of the text has been removed.

o pag 11 lines 374-376

The literature is placed after the specific inflammatory markers.

Point 7: the sentence on page 3 (lines 106-115) would be more appropriate for discussion

Response 7: In accordance with the comments, this sentence was moved to the discussion.

Point 9: It would be useful to provide (as Supplemental Material) the food frequency questionnaire, used to estimate the nutritional intake in T2D patients. This would allow to the reader to give a correct evaluation of the questionnaire.

Response 9: Thank Reviewer’s very much for your comments. According to Reviewer’s suggestion, the food frequency questionnaire (FFQ) translated from Polish to English, is included in Supplementary materials. Also, more details about the estimation dietary intakes of food components were included into the Materials and Methods paragraph in the new version of manuscript (p.5).

“FFQ enabled estimating the amount of individual products in patient diets during last year, and next based on these assessments the dietary intakes (in grams) were calculated.”

Points 10: please check the use of acronyms in the text, and use them where necessary in place of “Fatty acids” and “type 2 diabetes”

Response 10: As suggested, we have standarised all acronyms and abbreviations.

Points 11: Please, proofread and fix some typos, and grammatical mistakes (such as “veetables” Pag 6 line 246; or “at around” pag 10 line347

Response 11: Notice mistakes were corrected.

Point 12: Table 2 is not well aligned.

Response 12: Thank the Reviewer’s comment. In the amended version of manuscript is a new version of Table 2.

Round 2

Reviewer 1 Report

This reviewer would like to thank the authors to address the points presented in the previous report. There is a couple of concerns that this reviewer has in the revised manuscript. It is important to explicitly mention the limitations of the study in the manuscript as mentioned in the rebuttal responses 10 and 11. Another concern is related to the multilinear regression model. The authors state that all covariates play significant roles in determining the association between PUFA intake with HbA1c and FG. The authors should consider the interconnectivity of the covariates in each model. Age and gender in Model 2 can affect the level of physical activity, a covariate in Model 3. Physical activities can influence BMI and WHR, covariates in Model 4. The authors should consider merging all the adjusted models into one robust and predictive model.

Reviewer 2 Report

The manuscript has been carefully revised, and now it has definitely improved.